# Application of Variational Optical Flow Forecasting Technique Based on Precipitation Spectral Decomposition to Three Case Studies of Heavy Precipitation Events during Rainy Season in Hebei Province

**Jiyang Tian [1], Qingtai Qiu [2,*], Xiaoqi Zhao [2], Wenbin Mu [3], Xidong Cui [4], Chunqi Hu [4], Yajing Kang [5] and Yong Tu [1]**

[1] State Key Laboratory of Simulation and Regulation of Water Cycle in River Basin, China Institute of Water Resources and Hydropower Research, Beijing 100038, China; tjyshd@126.com (J.T.); tuyong@iwhr.com (Y.T.)

[2] College of Water Conservancy and Civil Engineering, Shandong Agricultural University, Tai'an 271018, China; 17866702767@163.com

[3] College of Water Resources, North China University of Water Resources and Electric Power, Zhengzhou 450000, China; muwenbin@ncwu.edu.cn

[4] Hebei Hydrological Survey Research Center, Shijiazhuang 050031, China; 13633188609@139.com (X.C.); heb_hcq@126.com (C.H.)

[5] China South-to-North Water Diversion Corporation Limited, Beijing 100071, China; kangyajing58@126.com

\* Correspondence: qqt31415926@163.com

**Abstract:** Short-term heavy precipitation is a crucial factor that triggers urban waterlogging and flash flood disasters, which impact human production and livelihood. Traditional short-term forecasting methods have time- and scale-based limitations. To achieve timely, location-specific, and quantitative precipitation forecasting, this study applies the precipitation spectral decomposition algorithm, along with variational echo tracking and autoregressive AR2 extrapolation techniques, to forecast three cases of heavy precipitation events during the rainy season in Hebei Province. The variational optical flow extrapolation forecasting based on precipitation spectral decomposition has a forecasting lead time of up to 3 h. However, noticeable discrepancies in forecast accuracy can be observed around 2 h, and the forecasting skill gradually weakens with longer lead times. For 3 h lead time forecasts, substantial variability occurs among different performance metrics, lacking clear comparability. The effective forecast lead time for variational optical flow forecasting based on precipitation spectral decomposition is up to 1.6 h for severe convective weather systems and up to 2.2 h for stratiform cloud weather systems. Overall, the forecast effect of this method is good in the three rainfalls—the highest CSI is up to 0.74, the highest POD is up to 0.87, and the forecast accuracy and success rate are high.

**Keywords:** nowcasting; variational echo tracking algorithm; precipitation spectral decomposition; AR2 autoregressive model

## 1. Introduction

The occurrence, evolution, and disappearance of precipitation in severe convective weather and precipitation caused by it are very fast, and its prediction and early warning are key and difficult points in the field of meteorology and hydrology. The quality of the observation data, the precipitation prediction method used, and the timeliness of the forecast period all influence the accuracy of its prediction. With the advancement of Doppler weather radar remote sensing technology, high-quality capture of rainfall spatial distribution information and rainfall inversion based on a volume scanning model provide basic data support for precipitation forecasting. Based on this, Browning proposed that precipitation approaching forecast is a forecast with high temporal and spatial resolution that the weather will change significantly in a short time (0~3 h) by radar echo extrapolation,

which has become one of the important researches in the hydrometeorological field [1,2]. According to Austin and Bellon, the approach prediction algorithm should include two components: echo identification and tracking and extrapolation prediction [3]. The optimal displacement is predicted after the echo is identified and the echo field is established. The approach prediction method based on weather radar echo tracking and extrapolation can show strong, good, and complete convective weather structure and convection movement, especially in areas with continuous radar reflectivity, which can better construct echo field, track, and extrapolate echo displacement changes.

At the moment, weather radar extrapolation approach prediction is divided into two categories: (1) single centroid tracking methods, such as Titan (Thunderstorm Identification, Tracking, Analysis, and Now Casting) and Scit (The Storm Cell Identification and Tracking), for identifying and tracking strong thunderstorm cells [4,5]. (2) Algorithms for tracking echo [6,7] for identifying and tracking large-scale precipitation areas, including trec-tracking radar echo by correlation and its derivatives, OF—Optical Flow, VET—Variational Echo Tracker, and so on [8,9]. Based on storm cell tracking and prediction, the centroid position of thunderstorms was identified, tracked, and predicted primarily using reflectivity data, and storm tracking and prediction were realized [10]. To identify and track the fusion and separation of convective cells, TITAN employs a combined optimization algorithm. It is impossible to distinguish storm clusters due to its single threshold for identifying cell movement and displacement changes. Based on mathematical morphology, Han et al. proposed ETITAN for storm identification. According to an application example, ETITAN's near success index (CSI) is 93% higher than TITAN's [11]. The historical trajectory of the storm is tracked according to the pixel or regional echo, and the regional tracking and forecasting is carried out by establishing an extrapolation model based on regional tracking and forecasting, such as the optical flow method [12]. Tuttle and colleagues considered the systematicity of radar echo on a large scale [13] and replaced TREC's backward extrapolation mode with the semi-Lagrangian advection scheme RPM-SL. This weakened the influence of the disordered vector caused by the excessive threshold in the echo field and made MTREC show good consistency and continuity in forecasting the rotation characteristics of precipitation [14]. The optical flow method, on the other hand, tracks pixels based on changes in image gray level, replaces the echo vector field with the calculated radar echo optical flow field, and analyzes the temporal and spatial changes of echo [15], making it suitable for strong convective precipitation systems and stratiform cloud precipitation systems.

However, the precipitation field contains many scales, and the influence of the thermodynamic environment on the change of echo intensity is not considered, resulting in a lack of forecasting ability of echo intensity change trend, which leads to different life span and predictability of precipitation at different scales [16]. The shorter the life span and the worse the predictability, the smaller the scale of precipitation. If you predict the entire scale of the precipitation field, the prediction error will be too large. The Horn–Schunk method uses global smoothing, while the Lucas–Kanade method uses local matching. In 1995, Laroche et al. proposed a Variational Echo Tracking (VET) algorithm [17]. On this foundation, McGill University in Canada created the MAPLE approach prediction system. Germann et al. used MAPLE to track and predict a 1424 h rainfall event in the continental United States, and the results showed that the average forecast time limit based on MAPLE was 5.1 h, which was clearly better than the forecasted effect of the Euler persistence algorithm [18]. Mandapara et al. forecasted 20 precipitation events that occurred in the Swiss alpine region from 2005 to 2010, and the results showed that the time limit of credible forecast based on MAPLE reached 3 h in the alpine region [19]. Lee et al. used MAPLE to forecast several summer precipitation events on the Korean Peninsula in 2008, and the results showed that the effective forecasting time was 2.5 h [20]. In comparison to the life of the precipitation model, the Lagrange advection scheme affects the radar precipitation model, which doubles the predictability of a given confidence level.

Given the limitations of traditional forecasting methods in tracking and forecasting, as well as the superiority of the VET algorithm, this study used previous studies, combined with the precipitation spectrum decomposition method [21], variational echo tracking technology, and the autoregressive AR2 model [22], to track and forecast three typical rainfalls in Hebei Province during the rainy season, and evaluated the effect of precipitation forecasting.

## 2. Data and Study Area

The data used in this study consist of three weather events observed by SA-band Doppler radar within the territory of Hebei Province, China. The radar base data were obtained from the China Meteorological Administration, in **.bin.bz2 format. As shown in Figure 1, the scanning radius of a single S-band radar was 250 km (with an effective scanning radius of 230 km), providing complete coverage of the study area. The radar operated with a scan completed every 6 min at nine different elevation angles, with a spatial resolution of 1 km × 1 km. The three weather events studied were three heavy rainfall events that occurred on 21 July 2012, 20 July 2016, and 18 July 2021 in Hebei Province. The rainfall event on 21 July 2012 was associated with convective weather systems, whereas the events on 20 July 2016 and 18 July 2021 were associated with stratiform cloud weather systems, the details are shown in Table 1.

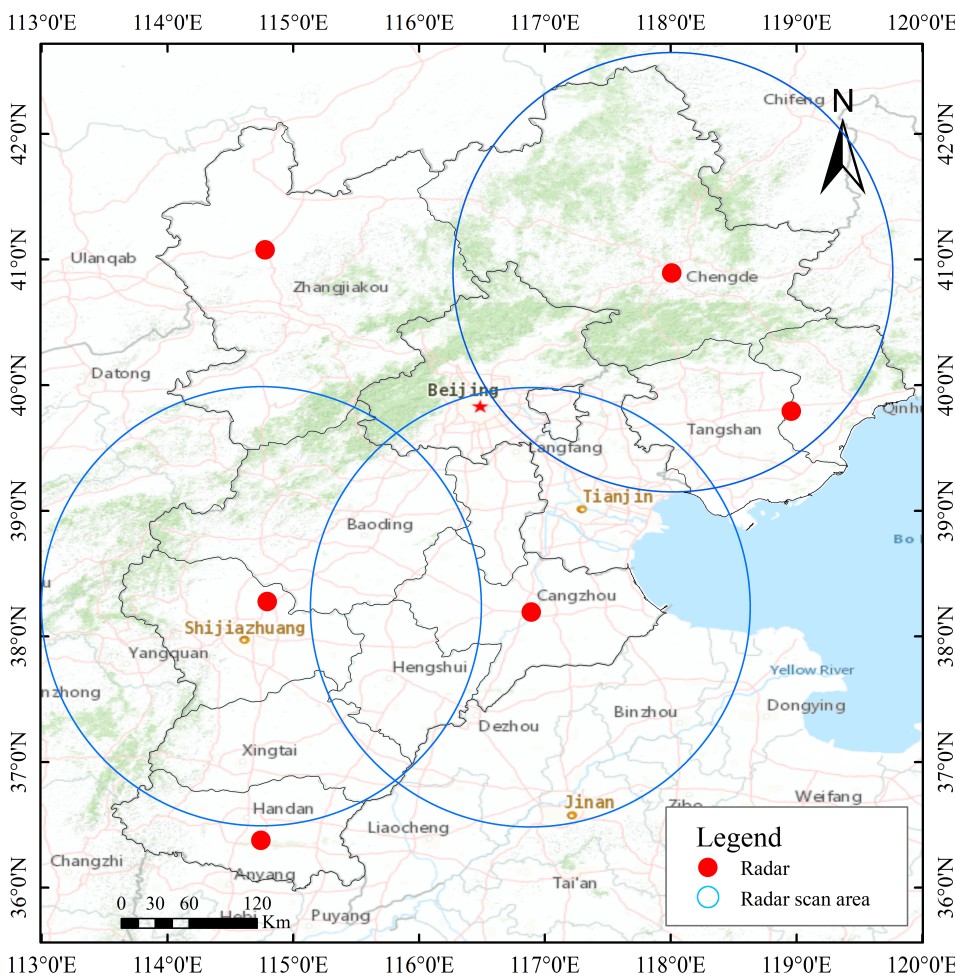

**Figure 1.** Distribution of Doppler weather radar stations in Hebei province.

**Table 1.** Summary of rainfall events.

| Rainfall Events | Rainfall Time | Weather System |
|:---:|:---:|:---:|
| Event I | 21 July 2012 from 9:00 to 21:00 | Convective Precipitation System |
| Event II | 20 July 2016 from 0:00 to 18:00 | Stratiform Cloud Precipitation System |
| Event III | 18 July 2021 from 0:00 to 11:00 | Stratiform Cloud Precipitation System |

## 3. Principles and Methods

### 3.1. Estimation of Advection Field

Nowcasting is based on the echoes or Quantitative Precipitation Estimation (QPE) of adjacent radar images, which determines the historical position of each grid and estimates the velocity vector of movement for each grid, forming an advection field. The Eulerian conservation algorithm is the simplest method for estimating advection field, assuming that echoes are stationary and considering the radar QPE at time t as the radar QPF at time t + 1, which has no practical significance in forecasting s [23] The Lagrangian conservation algorithm is currently the most commonly used method for advection estimation [24], Optical flow method is based on this principle for echo tracking [25], assuming that the total precipitation intensity does not change over time and can be described using the following formula:

$$\frac{dR}{dt} \equiv 0 \tag{1}$$

$R$ represents precipitation intensity or precipitation rate, in units of mm/h. Equation (1) can be further expressed as:

$$\frac{\partial R}{\partial x}\frac{\partial x}{\partial t} + \frac{\partial R}{\partial y}\frac{\partial y}{\partial t} + \frac{\partial R}{\partial t} = 0 \tag{2}$$

Or

$$u\frac{\partial R}{\partial x} + v\frac{\partial R}{\partial y} + \frac{\partial R}{\partial t} = 0 \tag{3}$$

Or

$$\mathbf{v}\cdot\nabla R + \dot{R} = 0 \tag{4}$$

where, $u, v$ are the components of precipitation velocity in the $x, y$ coordinates, respectively, with units of m/s; $\dot{R} = \frac{\partial R}{\partial t}$; and the velocity vector $\mathbf{v} = (u, v)$. The above equation is referred to as the Optical Flow equation.

In practical applications, the Optical Flow equation can be expressed as:

$$u\frac{\Delta R_n}{\Delta x} + v\frac{\Delta R_n}{\Delta y} + \frac{\Delta R_n}{\Delta t} = 0 \tag{5}$$

where $R_n$ represents the precipitation intensity at time step $n$, $u$ and $v$ are the components of the precipitation's velocity, $\frac{\Delta R_n}{\Delta x}$ and $\frac{\Delta R_n}{\Delta y}$ represent the rate of change of $R_n$ in the x and y directions, respectively, and $\frac{\Delta R_n}{\Delta t}$ represents the rate of change of $R_n$ at time t.

The optical flow equation involves two unknowns, $u, v$, which cannot be solved at a single grid point. In this study, the VET Ialgorithm is used for solving, which was initially developed for retrieving three-dimensional wind fields from single Doppler radar data were later applied to estimating advection fields for precipitation forecasting. VET searches for the optimal solution of $\mathbf{v}(u, v)$ over the selected spatial domain $\Omega$ by minimizing the cost function $J_{VET}(\mathbf{v})$, and the calculation formula is as follows [17]:

$$J_{VET}(\mathbf{v}) = J_Z + J_2 \tag{6}$$

The cost function $J_{VET}(v)$ consists of two components—$J_Z$ as the residual term of the optical flow equation and $J_2$ as the smoothing constraint term. They are defined as follows:

$$J_Z = \iint_{\Omega} \beta(\boldsymbol{x})[Z(t_0, \boldsymbol{x}) - Z(, \boldsymbol{x} - v\Delta t, y - v\Delta t, t_0 - \Delta t)]^2 dxdy \tag{7}$$

$$J_2 = \gamma \iint_{\Omega} \left(\frac{\partial^2 u}{\partial x^2}\right)^2 + \left(\frac{\partial^2 u}{\partial y^2}\right)^2 + 2\left(\frac{\partial^2 u}{\partial x \partial y}\right)^2 + \left(\frac{\partial^2 v}{\partial x^2}\right)^2 + \left(\frac{\partial^2 v}{\partial y^2}\right)^2 + 2\left(\frac{\partial^2 v}{\partial x \partial y}\right)^2 dxdy \tag{8}$$

where $Z$ represents the radar reflectivity factor in units of dB, which can also be expressed in decibels per unit intensity ($dBI$) using the conversion formula $dBI = 10log_{10}I$. $\beta(\boldsymbol{x})$ denotes the weight of data quality, and $\gamma$ is the weight of the smoothing constraint $J_2$, which is used to adjust the smoothness. To avoid convergence to local minima, VET employs a multi-scale analysis approach, starting with larger grid sizes and gradually searching for the global minimum of the cost function.

### 3.2. Precipitation Spectrum Decomposition Algorithm

Precipitation fields contain multiple scales, with different scales of precipitation having varying lifetimes and predictability. Small-scale precipitation tends to have short lifetimes and low predictability. To accurately describe the characteristics of precipitation at different scales, spectral decomposition algorithms are used to decompose precipitation or radar echoes, as shown in Figure 2. For example, a radar echo field of size $L \times L$ can be decomposed into layers of different scales, with each layer representing precipitation at a different scale. The total radar echo or precipitation was then obtained by summing up the layers (in decibels (dB)) or multiplying them (in linear units). The calculation formula is as follows [21].

$$dBZ_{i,j}(t) = \sum_{k=1}^{n} X_{k,i,j}(t), \ i = 1, \ldots, L, \ j = 1, \ldots, L, \ L = 2^n \tag{9}$$

where $dBZ_{i,j}(t)$ represents the reflectivity factor at time $t$ for position $(i, j)$, while $X_{k,i,j}(t)$ represents the reflectivity factor (unit: dB) at time $t$ for the $k$th scale at the position $(i, j)$.

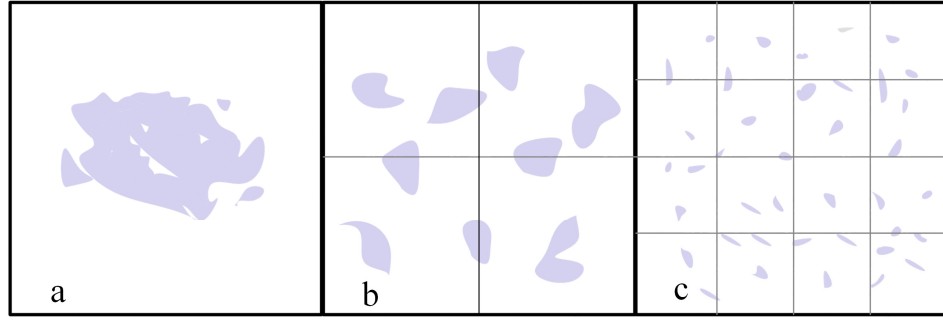

**Figure 2.** Schematic diagram of precipitation scale decomposition ((**a**) is radar QPE, (**b**,**c**) are precipitation fields of different scales after decomposition).

Once the precipitation field is decomposed into multiple scale layers, the precipitation at each layer and each time step was forecasted independently. The total forecasted precipitation at a given time step was obtained by summing the forecasted precipitation from different layers with weights that are determined based on the lifespan of precipitation at different scales. Precipitation with larger scales, which has longer lifespans, was assigned higher weights in the accumulation.

### 3.3. AR2 Autoregressive Extrapolation Forecasting

Once the advection field and the scale decomposition of the precipitation field are calculated, the next step is to perform extrapolation forecasting of precipitation. The basic

principle of extrapolation forecasting is to propagate the observed or analysed precipitation to the forecast time step using a forward or backward method based on the advection field [22]. In this study, an autoregressive (AR) scheme of forecasting model of order n was used, where n can be 1, 2, 3, and so on. Among them, the second-order AR model (AR2) performs the best and is commonly used. In AR2, for each decomposed precipitation layer k, two correlation coefficients, $\rho_{k,1}(t)$ and $\rho_{k,2}(t)$, are calculated. $\rho_{k,1}(t)$ represents the correlation between $Z_k(t-1)$ and $Z_k(t)$, where $Z_k(t-1)$ is shifted by $(\Delta x, \Delta y)$ distance based on the advection field. $\rho_{k,2}(t)$ represents the correlation between $Z_k(t-2)$, where $Z_k(t-2)$ is shifted by $(2\Delta x, 2\Delta y)$ distance based on the advection field. Similarly, $Z$ can also be $dBI$. The parameters $\varnothing_{k,1}(t)$ and $\varnothing\varnothing_{k,2}(t)$ of the AR2 model are established using $\rho_{k,1}(t)$ and $\rho_{k,2}(t)$, and the calculation formula is as follows [26].

$$\varnothing_{k,1}(t) = \frac{\rho_{k,1}(t)\left\{\rho_{k,1}(t)[1-\rho_{k,2}(t)]\right\}}{1-\rho_{k,1}(t)^2} \tag{10}$$

$$\varnothing_{k,2}(t) = \frac{\rho_{k,2}(t)-\rho_{k,1}(t)^2}{1-\rho_{k,1}(t)^2} \tag{11}$$

The forecast at time $t+1$ is given by:

$$Z_{k,i,j}(t+1) = \varnothing_{k,1}(t)Z_{k,i,j}(t) + \varnothing_{k,2}(t)Z_{k,i,j}(t-1) \tag{12}$$

More generally, the forecast at time $t+n+1$ is given by:

$$Z_{k,i,j}(t+n+1) = \varnothing_{k,1}(t)Z_{k,i,j}(t+n) + \varnothing_{k,2}(t)Z_{k,i,j}(t+n-1) \tag{13}$$

In the above equation, $Z_{k,i,j}(t+1)$ represents the extrapolation of the precipitation field for the $k$-th scale by $(\Delta x, \Delta y)$ at time $t+1$, while $Z_{k,i,j}(t+n+1)$ represents the extrapolation of the precipitation field for the $k$-th scale by $(n\Delta x, n\Delta y)$ at time $t+n+1$. The final output of the forecast field was obtained by accumulating the forecasts for $k$ different scales at the same forecast time.

$$Z_{i,j}(t+n+1) = \sum_k Z_{k,i,j}(t+n+1) \tag{14}$$

As small-scale precipitation tends to gradually diminish with increasing forecast lead time, resulting in a more uniform precipitation forecast field, it is necessary to correct for energy loss. One approach is to adjust the forecast field so that the proportion of forecast precipitation area exceeding a certain threshold is equivalent to the proportion of original reflectivity. For example, using a threshold of 15 $dBZ$, areas with reflectivity exceeding this threshold are considered as precipitation, whereas areas with reflectivity below the threshold are considered as no precipitation. The proportion of observed reflectivity area exceeding 15 $dBZ$, denoted as $f_{15}$, is calculated. Then, the reflectivity value in the forecast field corresponding to the same proportion, denoted as $Z_f$, is calculated. Because the forecast field tends to be more uniform, $Z_f$ is always smaller than or equal to 15 $dBZ$. Finally, the forecast field is corrected as follows:

$$dBZ_{i,j} = \begin{cases} dBZ_{i,j} + \left(15 - Z_f\right) \text{ If } dBZ_{i,j} > Z_f \\ \qquad\qquad 0 \text{ Otherwise} \end{cases} \tag{15}$$

where, $(i,j)$ represents the grid point location.

### 3.4. Evaluation Methods

In hydrometeorology, commonly used evaluation metrics include probability of detection (POD), false alarm ratio (FAR), and critical success index (CSI) [27]. These metrics allow for qualitative and quantitative evaluation of forecast results while considering spatial and temporal scales. The aforementioned metrics can be obtained through the evaluation method of contingency tables [28]. The evaluation of meteorological indicators needs to set

a threshold TR, judge the relationship between reflectance factor and TR, and then obtain the size of the evaluation value, as shown in Table 2.

**Table 2.** Qualitative evaluation table for binary rain events.

| Forecast Value | Observed Value | | Total |
| --- | --- | --- | --- |
| | **>TR** | **<TR** | |
| >TR | NA | NB | NA + NB |
| <TR | NC | ND | NC + ND |
| Total | NA + NC | NB + ND | |

In the context of this study, TR is used as the threshold for determining rainfall occurrence, where values greater than TR indicate rainfall. NA represents cases where both forecasted rainfall and observed radar rainfall occur simultaneously; NC represents cases where radar observes actual rainfall, but no rainfall is forecasted; NB represents cases where radar does not observe actual rainfall, but rainfall is forecasted; ND represents cases where radar does not observe rainfall and no rainfall is forecasted by the model. The definitions of detection rate, false alarm rate, critical success index, and root mean square error are as follows:

(1) Probability of Detection (*POD*): It describes the proportion of observed rainfall events that are correctly forecasted as exceeding the TR, which is:

$$POD = \frac{1}{N}\sum_{i=1}^{N}\frac{NA_i}{NA_i + NC_i} \tag{16}$$

(2) False Alarm Ratio (*FAR*): It describes the proportion of forecasted rainfall events that exceed the TR but are false alarms, which is,

$$FAR = \frac{1}{N}\sum_{i=1}^{N}\frac{NB_i}{NA_i + NB_i} \tag{17}$$

(3) *CSI*: In contrast to *POD*, this index measures the proportion of correct forecasts out of all possible occurrences of rainfall events, which is:

$$CSI = \frac{1}{N}\sum_{i=1}^{N}\frac{NA_i}{NA_i + NB_i + NC_i} \tag{18}$$

(4) Root Mean Square Error (*RMSE*)

$$RMSE = \sqrt{\frac{1}{M}\sum_{j=1}^{M}(P_j - Q_j)^2} \tag{19}$$

where $P_j$ represents the simulated rainfall amount at a particular grid point, $Q_j$ represents the observed cumulative rainfall amount over the entire observation period at the same grid point, and $M$ represents the total number of grid points.

## 4. Forecast Results

As stated in Principles and Methods, this paper establishes the balance relationship of reflectivity factors of adjacent radar images and obtains the initial echo motion vector field using the variational echo tracing principle and the Lagrange conservation law. We analyzed the information characteristics of radar echo images, realized the scale decomposition of the precipitation field, and forecasted precipitation independently at each moment. The autoregressive AR2 model was used to forecast extrapolation, and precipitation was

superimposed according to scale, resulting in the extrapolation forecast of three selected rainfall events. Figure 3 compares the application of the method to the 0~3 h forecast results from 20:00 on 21 July 2012. Taking the example of 22 May 2021 at 12:24, as shown in Figure 4 there was no significant change in radar QPE from 12:54 to 13:54. Precipitation is distributed in a band from southwest to northeast. The precipitation began to weaken and move south from 13:54. By 14:24, while the precipitation in the north-eastern region had mostly disappeared, the precipitation in the southern region continued to intensify. Over the 3 h period, the radar QPE showed a pattern of decreasing precipitation in the north-eastern region and increasing precipitation in the southern region, with a tendency of southward movement and scattered precipitation areas. The spatial distribution and intensity of the radar quantitative precipitation forecast (QPF) during 12:54 to 13:24 were similar to the QPE. From 14:24 to 15:24, while the precipitation in the southward region gradually weakened and almost disappeared by 14:54, the precipitation in the north-eastern region gradually intensified, with no significant changes in spatial distribution. The 3 h radar QPF showed a trend of decreasing precipitation in the southward region and increasing precipitation in the north-eastern region, with precipitation becoming more concentrated in small areas and no scattered precipitation areas. To objectively evaluate the forecasting results, this study conducted evaluations of forecast performance in terms of spatial and temporal scales. The specific evaluation results are as follows:

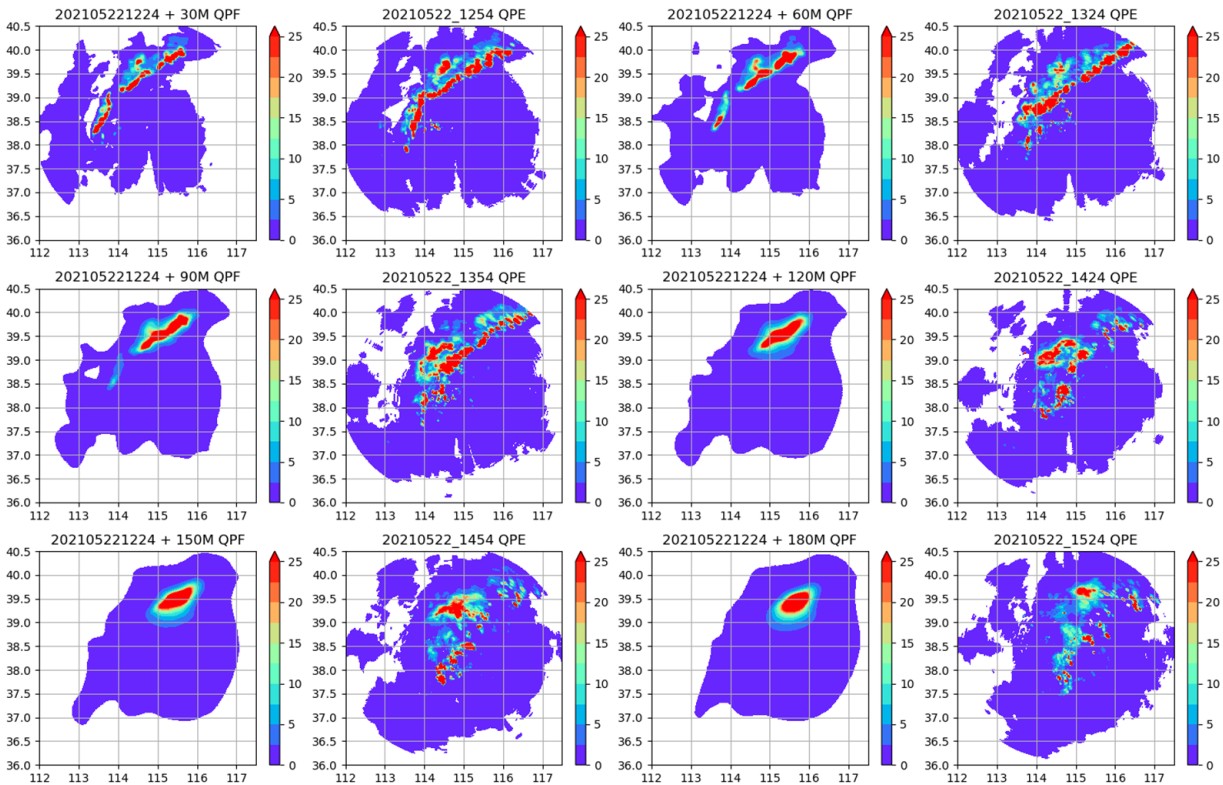

**Figure 3.** Comparison between radar quantitative precipitation estimation (QPE) and radar quantitative precipitation forecast (QPF).

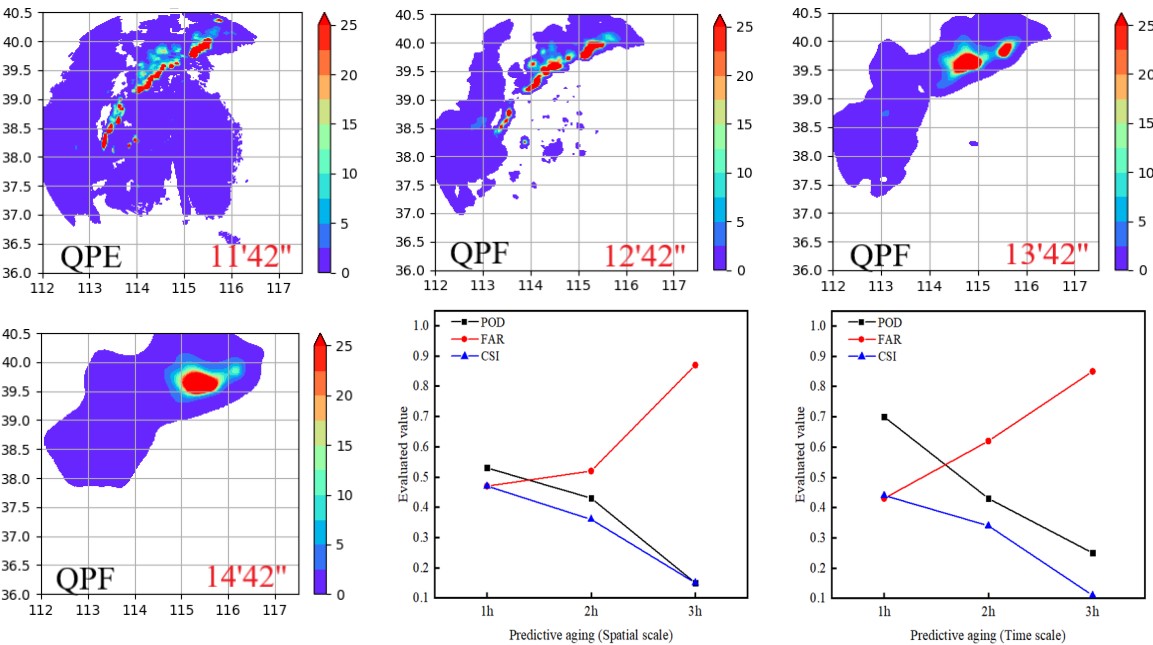

**Figure 4.** Trend chart of time-scale evaluation indices for Event I.

### 4.1. Spatial-Scale Evaluation

As Table 3 shows, the data indicate that the variational optical flow extrapolation technique based on precipitation spectral decomposition has a lead time of no more than 3 h, with deviations becoming noticeable around 2 h. For a 1 h lead time forecast, the accuracy measure POD ranges from 0.5 to 0.9, the FAR ranges from 0.3 to 0.5, the CSI ranges from 0.4 to 0.8, and RMSE ranges from 1.1 to 1.6. The maximum value of POD, 0.87, occurs in event II, and the minimum value of FAR, 0.31, occurs in event II as well. The maximum value of CSI, 0.74, also occurs in event II, while the minimum value of RMSE, 1.13, occurs in event II. For a 2 h lead time forecast, the range of POD is between 0.4 and 0.7, FAR ranges from 0.4 to 0.6, CSI ranges from 0.3 to 0.6, and RMSE ranges from 1.9 to 2.2. The maximum value of POD, 0.62, occurs in event II, and the minimum value of FAR, 0.49, occurs in event II as well. The maximum value of CSI, 0.54, also occurs in event II, while the minimum value of RMSE, 1.89, occurs in event II. For a 3 h lead time forecast, there are significant differences among the various indicators, and they are not easily comparable. Based on the forecast results, event II has the best forecast performance within the lead time, clearly outperforming event I and event III.

**Table 3.** Spatial scale evaluation results of rainfall forecast based on radar QPE.

| Event | Forecasting Lead Time | POD | FAR | CSI | RMSE | CSI/RMSE |
|---|---|---|---|---|---|---|
| | 1 h | 0.53 | 0.47 | 0.47 | 1.31 | 0.36 |
| I | 2 h | 0.43 | 0.52 | 0.36 | 1.92 | 0.19 |
| | 3 h | 0.15 | 0.87 | 0.15 | 3.57 | 0.04 |
| | 1 h | 0.87 | 0.31 | 0.74 | 1.13 | 0.65 |
| II | 2 h | 0.62 | 0.49 | 0.54 | 1.89 | 0.29 |
| | 3 h | 0.37 | 0.75 | 0.27 | 2.76 | 0.10 |
| | 1 h | 0.67 | 0.33 | 0.61 | 1.54 | 0.40 |
| III | 2 h | 0.49 | 0.50 | 0.43 | 2.18 | 0.20 |
| | 3 h | 0.32 | 0.78 | 0.24 | 3.10 | 0.08 |

*4.2. Temporal-Scale Evaluation*

As Table 4 shows, similar to the spatial scale evaluation results, at different time scales, the forecast performance is the best at a 1 h lead time, with a rapid decrease in forecast skill as the lead time increases. For a 1 h lead time forecast, the range of accuracy indicator POD is between 0.70 and 0.72, the range of false alarm rate indicator FAR is between 0.31 and 0.43, the range of critical success index CSI is between 0.44 and 0.61, and the range of RMSE is between 1.03 and 1.59. The maximum value of POD is 0.72, observed in rainfall event II, and the minimum value of FAR is 0.31, observed in rainfall event II. The maximum value of CSI is 0.61, observed in rainfall event II, and the minimum value of RMSE is 1.03, observed in rainfall event II. For a 2 h lead time forecast, the range of POD is 0.43–0.54, FAR is 0.60–0.62, CSI is 0.34–0.42, and RMSE is 1.7–1.95. The maximum value of POD, 0.54, occurs in event II, whereas the minimum value of FAR, 0.60, occurs in event II as well. The maximum value of CSI, 0.42, also occurs in event II, whereas the minimum value of RMSE, 1.76, occurs in event II. Similarly, for forecast lead times of 3 h, a significant disparity exists among various extrapolation forecast indicators, making it difficult to draw meaningful comparisons. Therefore, a detailed description is not provided here. Similar to the spatial scale, rainfall event II exhibits the best forecast performance within the lead time, clearly superior to rainfall events I and III.

**Table 4.** Time-scale evaluation results of three precipitation forecasts based on radar QPE.

| Event | Forecasting Lead Time | POD | FAR | CSI | RMSE | CSI/RMSE |
|---|---|---|---|---|---|---|
| | 1 h | 0.7 | 0.43 | 0.44 | 1.17 | 0.38 |
| I | 2 h | 0.43 | 0.62 | 0.34 | 1.89 | 0.18 |
| | 3 h | 0.25 | 0.85 | 0.11 | 3.7 | 0.03 |
| | 1 h | 0.72 | 0.31 | 0.61 | 1.03 | 0.59 |
| II | 2 h | 0.54 | 0.60 | 0.42 | 1.76 | 0.14 |
| | 3 h | 0.21 | 0.86 | 0.25 | 2.91 | 0.01 |
| | 1 h | 0.72 | 0.41 | 0.54 | 1.59 | 0.34 |
| III | 2 h | 0.51 | 0.61 | 0.35 | 1.95 | 0.15 |
| | 3 h | 0.26 | 0.83 | 0.21 | 3.12 | 0.07 |

For precipitation event I, the POD evaluation index shows a sharp decrease between 2 to 3 h, while the FAR shows a sharp increase, with larger changes compared to 1 to 2 h. The CSI starts to be below 0.3 at 1.6 h and continues to decline, as shown in Figure 4. For precipitation event II, the POD and CSI indices show relatively gradual declines with increasing forecast lead time from 0 to 3 h, while the FAR shows a gradual increase. The CSI starts to be below 0.3 at 2.8 h, as shown in Figure 5. For precipitation event III, the FAR shows a more gradual change between 2 to 3 h compared to 1 to 2 h, and the POD and CSI show gradual declines with increasing forecast lead time. The CSI starts to be below 0.3 at 2.6 h, as shown in Figure 6. In terms of spatial scale, the precipitation forecast performance of precipitation event II is better than that of precipitation event III, and the forecast performance of stratiform cloud precipitation systems is better than that of severe convective precipitation systems.

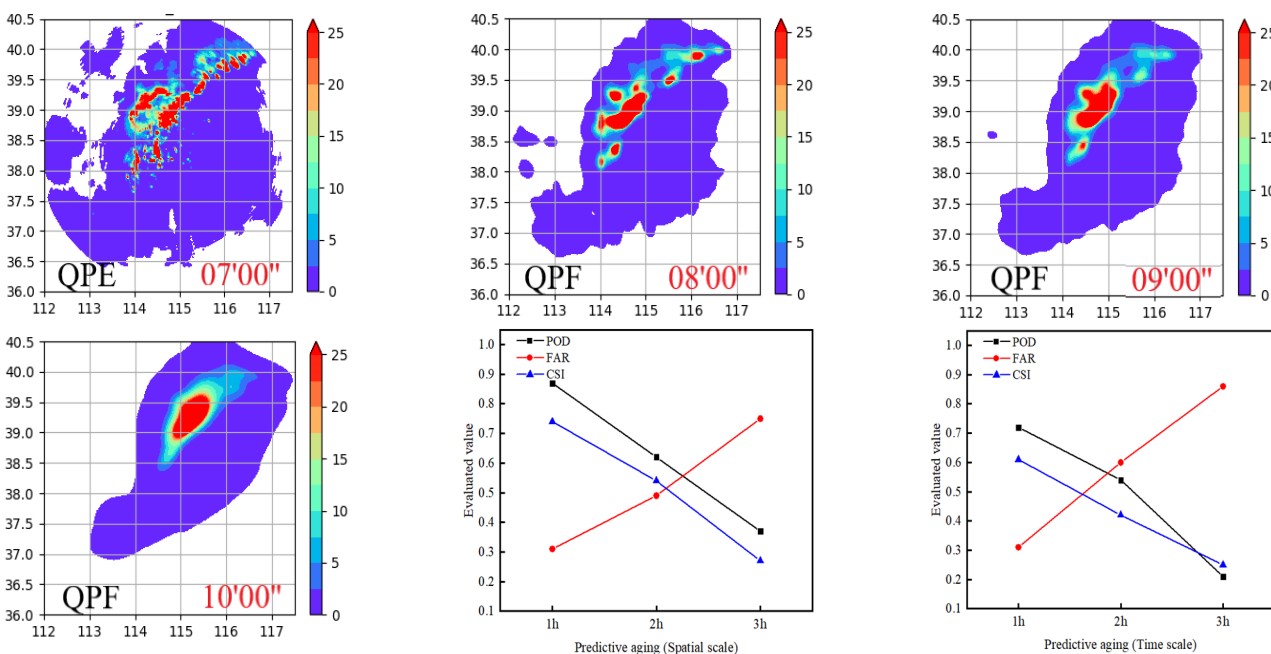

**Figure 5.** Trend chart of time-scale evaluation indices for Event I–Event II.

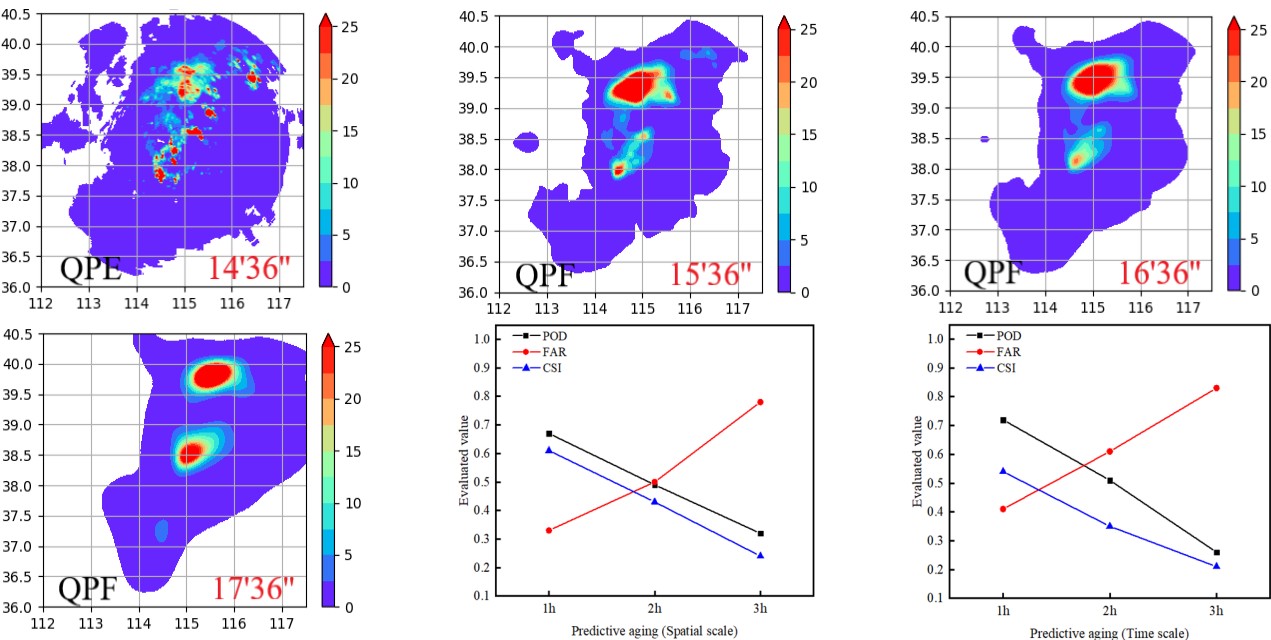

**Figure 6.** Trend chart of time-scale evaluation indices for Event I–Event III.

In the evaluation of precipitation for event I, the POD shows a sharp decrease at 2–3 h, whereas FAR exhibits a gradual increasing trend. CSI shows a gradual decrease, dropping below 0.3 starting from 2.2 h and continuing to decline, as shown in Figure 6. For event II, the decrease in POD at 2–3 h is larger compared to that at 1–2 h, whereas FAR shows a gradual increasing trend, and CSI shows a gradual decreasing trend. CSI drops below 0.3 starting from 2.7 h and continues to decline, as shown in Figure 7. For event III, the changes in precipitation evaluation indicators are relatively modest within the 0–3 h forecast period. POD and CSI show a gradual decrease with increasing forecast lead time, whereas FAR shows a gradual increase. CSI drops below 0.3 starting from 2.4 h, as shown in Figure 6.

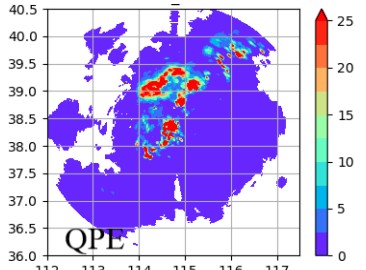 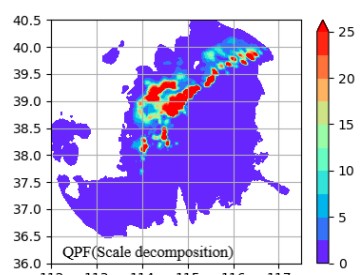 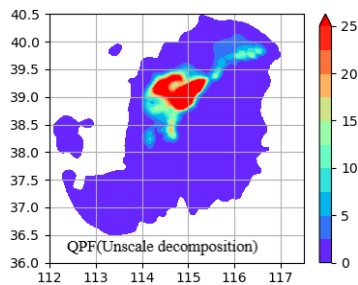

**Figure 7.** Comparison of radar quantitative precipitation estimate (QPE), scale-decomposed precipitation forecast quantitative precipitation forecast (QPF), and non-scale-decomposed precipitation forecast QPF.

In terms of forecast lead time, the precipitation forecast performance of event II is better than that of event III, and the forecast performance of stratiform cloud precipitation system is better than that of severe convective precipitation system.

## 5. Discussion

The forecast performance of optical flow and cross-correlation methods has revealed several limitations in special weather conditions. The optical flow method does not account for the rotational characteristics of echoes in tropical typhoon systems, resulting in significant deviations between the forecast results and actual observations [29] However, the cross-correlation method does not consider the vertical motion of echoes, which can lead to tracking failures in severe convective precipitation systems due to the intense motion of echoes [30].

Compared to optical flow and cross-correlation methods, the variational extrapolation method based on precipitation spectrum decomposition offers more precise tracking of echoes and higher accuracy in extrapolation forecasting. This is specifically manifested in the following aspects: the precipitation spectrum decomposition method decomposes echoes into different scales, avoiding the use of a homogenized echo field for extrapolation, which effectively mitigates forecast errors caused by short lifetime of small-scale precipitation, as shown in Figure 7; the variational echo tracking method based on differential images of consecutive sequence images provides high forecasting accuracy and is suitable for short-term forecasting under complex weather background conditions. The autoregressive AR2 model extrapolates forecasts based on the reflectivity factors of three consecutive time steps, instead of using a single linear extrapolation method, resulting in convincing extrapolation results. The forecast results indicate that this method exhibits high accuracy in the 0–2 h forecast. However, the accuracy gradually decreases in the 2–3 h forecast, with considerable deviations from actual observations in the 2.5–3 h forecast, indicating that certain limitations still occur in this method. Further discussion is provided below.

The motion field is generated from consecutive radar echo images, and the Lagrangian conservation algorithm assumes that the motion speed of precipitation echoes remains constant in a short period of time [31] However, in actual meteorological environments, the speed of echo motion can change with weather conditions, as shown in Figure 8. As the forecast lead time extends, a significant difference occurs between the actual motion field speed and the Lagrangian constant speed, which is an important factor causing forecast errors. By incorporating forecast information, the issue of short-term forecast errors caused by constant speed can be addressed effectively [32]. Although this method has idealized assumptions, its prediction success rate is relatively high, for example, compared with the Kanade–Lucas–Tomasi optical flow method, CSI was 0.2 higher [33]. Compared with the cross-correlation method used to forecast a local heavy precipitation event in Hong Kong, the FAR decreased by about 10% [34]. The precipitation spectral decomposition algorithm decomposes precipitation or echo into different components, and the total forecasted precipitation at a given time was obtained by summing the forecasted precipitation from

different layers with weighted accumulation. However, precipitation is a complex nonlinear process, and the physical mechanisms of its generation, evolution, and dissipation are still under further research [35]. In the process of weight allocation, relying solely on the influence factor of precipitation lifetime to assign different weights to precipitation may result in inaccurate forecasts. Therefore, when assigning weights to precipitation of different scales, factors such as the causation of precipitation, precipitation types, and their evolution mechanisms should be considered, and key factors affecting their variations should be identified for reasonable weight allocation [36]. Currently, the parameters of the autoregressive AR2 model are determined using empirical formulas, and dynamically determining the model parameters based on the latest parameters may further improve its performance [34].

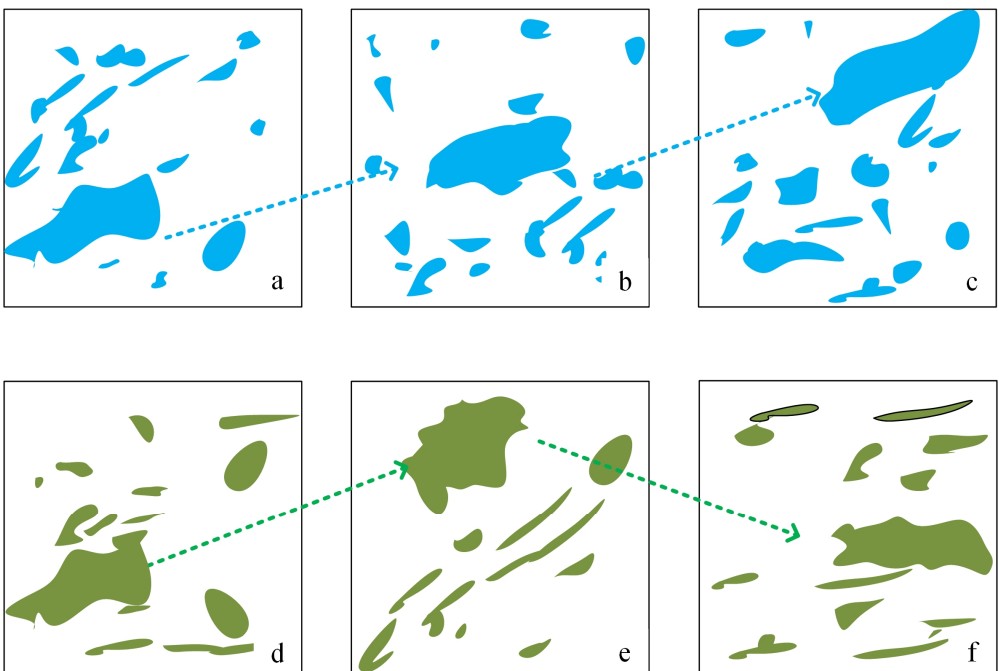

**Figure 8.** Comparison of echo motion vectors between the forecasted precipitation field and actual precipitation field at the same time ((**a**,**d**) are the initial radar field QPE, (**b**,**c**) are the motion track diagram of the forecast field echo, (**e**,**f**) are the motion track diagram of the actual precipitation field echo).

The integration of forecasts is the current main trend in nowcasting. The development of new approaches that combine spectral decomposition algorithms with numerical modelling techniques, such as variational echo tracking, is crucial in achieving more accurate forecasts. Additionally, it is important to fully consider a series of physical mechanisms related to precipitation, such as its generation, evolution, and dissipation, and comprehensively evaluate the impact of precipitation influencing factors on forecast results. Furthermore, adopting optimal parameter fitting schemes to determine model parameters can help reduce forecast errors introduced by the model. The variational echo extrapolation method based on precipitation spectral decomposition has shown some advantages in tracking and forecasting, but further improvements are still needed. Building on the findings of this study, efforts are being made to propose more accurate short-term forecasting methods and establish a more universally applicable short-term forecasting system.

## 6. Conclusions

In this study, Doppler weather radar data from three heavy rainfall events in Hebei Province were utilized to construct a nowcasting system based on precipitation spectral decomposition, variational echo tracking technology, and AR2 autoregressive model extrapolation technique. The forecasting performance of precipitation was evaluated in terms of spatial and temporal scales. The main conclusions are as follows:

(1) The variational optical flow extrapolation forecast based on precipitation spectral decomposition typically has a forecast lead time of no more than 3 h. The deviation becomes noticeable after approximately 2 h, and as the forecast lead time extends, the forecasting ability gradually decreases. For forecasts with a lead time of 3 h, significant differences occur among various indicators, and clear comparability is lacking.

(2) The forecast performance of stratiform cloud weather systems is superior to that of severe convective weather systems, as demonstrated by the following: On the spatial scale, the CSI for precipitation in event I begins to decrease below 0.3 after 1.6 h and continues to decline, whereas the CSI for precipitation in event II starts to fall below 0.3 after 2.8 h. Compared with the optical flow method, the effective prediction time is extended by 0.4 h. On the temporal scale, the CSI for precipitation in event I begins to decrease below 0.3 after 2.2 h and continues to decline, whereas the CSI for precipitation in event II starts to fall below 0.3 after 2.7 h, and for event II, it starts to fall below 0.3 after 2.4 h.

(3) The effective forecast lead time of the variational optical flow prediction technique based on precipitation spectrum decomposition is 1.6 h under severe convective weather conditions, and 2.2 h under stratiform cloud weather conditions. Overall, the forecast effect of this method is good in the three rainfalls, the highest CSI is up to 0.74, the highest POD is up to 0.87, and the forecast accuracy and success rate is high, but there are still some deviations. The research will continue in the later period and strive to find a better forecasting method.

**Author Contributions:** Conceptualization, Q.Q. and J.T.; methodology, J.T.; software, W.M.; validation, W.M., X.Z. and Y.T.; investigation, X.C. and Y.K.; resources, C.H.; data curation, C.H.; writing—original draft preparation, Q.Q. and X.Z.; writing—review and editing, J.T.; visualization, J.T.; supervision, X.C.; funding acquisition, Q.Q. and J.T. All authors have read and agreed to the published version of the manuscript.

**Funding:** This research was funded by the National Natural Science Foundation of China, grant number 51909274; IWHR Research & Development Support Program, grant number JZ0145B032020 and the Open Research Fund of State Key Laboratory of Simulation and Regulation of Water Cycle in River Basin (China Institute of Water Resources and Hy-dropower Research), grant number IWHR-SKL-KF202118.

**Data Availability Statement:** The datasets used and/or analyzed during the current study are available from the corresponding author upon reasonable request.

**Conflicts of Interest:** The authors declare no conflict of interest.

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
