# Peer review of "Application of Variational Optical Flow Forecasting Technique Based on Precipitation Spectral Decomposition to Three Case Studies of Heavy Precipitation Events during Rainy Season in Hebei Province"

_water, doi:10.3390/w15122204_

Round 1
Reviewer 1 Report
The manuscript Application of Variational Optical Flow Forecasting Technique Based on Precipitation Spectral Decomposition during Rainy Season in Hebei Province applies the precipitation spectral decomposition algorithm, along with variational echo tracking and autoregressive AR2 extrapolation techniques, to forecast three cases of heavy precipitation events during the rainy season in Hebei Province. The paper is well written, except a few typo, I have no further comments.
1. Line 83, what does "1424-h" mean?
2. Line 129, the equation missing the right part, an "0"?
3. Line 151, Eq 3.7. Is it missing the y component?
4. Line 243, please explain "TR".
5. Line 260 to 283. The time format like 12:24, should be 12'34''.
Finally, the results of the method is expected that the Lagrangian conservation assumption that total precipitation intensity does not change over time limits its prediction lead time.
Author Response
Application of Variational Optical Flow Forecasting Technique Based on Precipitation Spectral Decomposition to Three Case Studies of Heavy Precipitation Events during Rainy Season in Hebei Province
Reply to Reviewer #1
We appreciate the constructive comments from the reviewer. We have modified according to the reviewer's opinion. Thank you for your advice.
Point 1: Line 83, what does "1424-h" mean?
Reply: The paper cited here is “Predictability of Precipitation from Continental Radar Images, Part IV: Limits to Prediction”. And “1424h” refers to 1424h rainfall. As shown in the picture.
Reference:
Germann U, Zawadzki I, Turner B. Predictability of precipitation from continental radar images. Part IV: Limits to prediction[J]. Journal of the Atmospheric Sciences, 2006, 63(8): 2092-2108.
Point 2: Line 129, the equation missing the right part, an "0"?
Reply: Formula input error, has been modified according to the modification suggestions, thank you.
Point 3: Line 151, Eq 3.7. Is it missing the y component?
Reply: Formula input Thank you for your suggestion. The input of the formula here is wrong and y variable is missing. Now it has been modified into the correct formula.error, has been modified according to the modification suggestions, thank you
Point 4: Line 243, please explain "TR".
Reply: In previous versions, DBZ was written in Table 2, while TR was used for the following description of the formula. In fact, DBZ and TR mean the same thing, both referring to the precipitation threshold. At present, all DBZ in Table 2 has been changed into TR, which is my negligence. Thank you for your opinion.
Point 5: Line 260 to 283. The time format like 12:24, should be 12'34''.
Reply: Thank you for your suggestion. Now the time format has been standardized and modified.
Point 6: Finally, the results of the method is expected that the Lagrangian conservation assumption that total precipitation intensity does not change over time limits its prediction lead time.
Reply:Yes, Lagrange's persistence equation is relatively ideal in the prediction process, which will have certain influence on the forecast results, which has been described in the discussion chapter of this paper. In the later stage, the optimization of Lagrange algorithm in the forecasting process will be carried out to achieve better forecast results. In addition, there are many references about Lagrange algorithm, which introduce the principle of Lagrange and its application in many fields in detail. On the whole, Lagrange algorithm establishes the relation of reflectance factors and realizes the complete prediction process, but its shortcomings need to be corrected in later study.
Reference:
Reich S. An explicit and conservative remapping strategy for semi‐Lagrangian advection[J]. Atmospheric Science Letters, 2007, 8(2): 58-63.
Staniforth A, Côté J. Semi-Lagrangian integration schemes for atmospheric models—A review[J]. Monthly weather review, 1991, 119(9): 2206-2223.
Conclusion words:
We appreciate the constructive comments and valuable suggestions from the reviewer. The manuscript has been carefully revised accordingly. Detailed explanations and discussions are given regarding these issues. In our revision, spelling errors are corrected and efforts are also made to improve the readability of the paper.

Reviewer 2 Report
1. The quality of the figures in this manuscript must be improved. Please see the details. The resolution of figures is low for figures 2, 3, 4, and 10. The resolution of the x-axis and y-axis in Figures 6,7,8, and 9 is low. Please add axis titles for all the figures. Please add subtitles in Figure 10.
2. It is hard to understand figures 1, 2, 3, and 4 since these figures are not mentioned anywhere in the text.
3. Generally, the reference number should be at the end of the sentence.
4. The introduction is not well structured. Please add the descriptions for the main work of each section in the introduction.
5. What is the contribution and novelty of this study? Please clarify them in the introduction and conclusion.
6. Section 4 Forecasting results is not well written, and it is hard to follow.
7. In lines 459-461 in Conclusion, the authors gave the statement “Overall, this method demonstrates good applicability in the nowcasting of typical precipitation events during the rainy season in Hebei Province. “However, this paper only used three historical heavy rainfall events, and this may be a limitation.
8. English writing must be improved.
1. Section 4 Forecasting results is not well written, and it is hard to follow.
2. English writing must be improved.
Author Response
Application of Variational Optical Flow Forecasting Technique Based on Precipitation Spectral Decomposition to Three Case Studies of Heavy Precipitation Events during Rainy Season in Hebei Province
Reply to Reviewer #2
We appreciate the constructive comments from the reviewer. We have modified according to the reviewer's opinion. Thank you for your advice.
Point 1: The quality of the figures in this manuscript must be improved. Please see the details. The resolution of figures is low for figures 2, 3, 4, and 10. The resolution of the x-axis and y-axis in Figures 6,7,8, and 9 is low. Please add axis titles for all the figures. Please add subtitles in Figure 10.
Reply: Thank you for your suggestion. The image does have the problem of low resolution. Now the resolution of the image has been improved according to the modification suggestions. The resolution of FIG. 2, 4 and 10 has been improved. Considering that the watershed map in FIG. 3 is not relevant to the paper, it has been deleted. The X - and Y-axis resolutions of Figures 6, 7, 8, and 9 have been improved. Figure 10 has subtitles added.
Point 2: It is hard to understand figures 1, 2, 3, and 4 since these figures are not mentioned anywhere in the text.
Reply: Thank you for your advice. According to the modification suggestions, the explanation of images was added in the article, so that readers could better read the article combined with images. This is indeed our negligence. Thank you again for your opinion.
Point 3: Generally, the reference number should be at the end of the sentence.
Reply: Thank you for your guidance. According to the revision opinions, the current reference numbers are all put at the end of the sentence, which is our negligence. Thank you again for your suggestion.
Point 4: The introduction is not well structured. Please add the descriptions for the main work of each section in the introduction.
Reply: Thank you for your advice. After receiving your letter, I have carefully read the introduction for several times. The organizational structure is really not clear and the explanation of each part of the work is missing. According to the modification opinions, the organization structure of the introduction part has been reinterpreted and the main work has been elaborated.
“The occurrence, evolution, and disappearance of precipitation in severe convective weather and precipitation caused by it are very fast, and its prediction and early warning are key and difficult points in the field of meteorology and hydrology. The quality of the observation data, the precipitation prediction method used, and the timeliness of the forecast period all influence the accuracy of its prediction. With the advancement of Doppler weather radar remote sensing technology, high-quality cap-ture of rainfall spatial distribution information and rainfall inversion based on volume scanning model provide basic data support for precipitation forecasting. Based on this, Browning proposed that precipitation approaching forecast is a forecast with high temporal and spatial resolution that the weather will change significantly in a short time (0~3h) by radar echo extrapolation, which has become one of the important researches in the hydrometeorological field [1] [2]. According to Austin and Bellon, the approach prediction algorithm should include two components: echo identification and tracking and extrapolation prediction[3]. The optimal displacement is predicted after the echo is identified and the echo field is established. The approach prediction method based on weather radar echo tracking and extrapolation can show strong, good, and complete convective weather structure and convection movement, especially in areas with con-tinuous radar reflectivity, which can better construct echo field, track, and extrapolate echo displacement changes.
At the moment, weather radar extrapolation approach prediction is divided into two categories: (1)single centroid tracking methods, such as Titan (Thunderstorm Identification, Tracking, Analysis, and Now Casting) and Scit (The Storm Cell Identifi-cation and Tracking), for identifying and tracking strong thunderstorm cells [4,5]. (2) Algorithms for tracking echo[6,7]for identifying and tracking large-scale precipitation areas, including trec-tracking radar echo by correlation and its derivatives, OF-Optical Flow, VET-Variational Echo Tracker, and so on.[8,9]. Based on storm cell tracking and prediction, the centroid position of thunderstorms is identified, tracked, and predicted primarily using reflectivity data, and storm tracking and prediction is realized [10]. To identify and track the fusion and separation of convective cells, TITAN employs a combined optimization algorithm. It is impossible to distinguish storm clusters due to its single threshold for identifying cell movement and displacement changes. Based on mathematical morphology, Han et al. proposed ETITAN for storm identification. Ac-cording to an application example, ETITAN's near success index (CSI) is 93% higher than TITAN's [11]. The historical trajectory of the storm is tracked according to the pixel or regional echo, and the regional tracking and forecasting is carried out by establishing an extrapolation model based on regional tracking and forecasting, such as the optical flow method [12]. Tuttle and colleagues considered the systematicity of radar echo on a large scale [13], and replaced TREC's backward extrapolation mode with the semi-Lagrangian advection scheme RPM-SL, which weakened the influence of the dis-ordered vector caused by the excessive threshold in the echo field, and made MTREC show good consistency and continuity in forecasting the rotation characteristics of precipitation [14]. The optical flow method, on the other hand, tracks pixels based on changes in image gray level, replaces the echo vector field with the calculated radar echo optical flow field, and analyzes the temporal and spatial changes of echo [15], making it suitable for strong convective precipitation systems and stratiform cloud precipitation systems.
However, the precipitation field contains many scales, and the influence of the thermodynamic environment on the change of echo intensity is not considered, re-sulting in a lack of forecasting ability of echo intensity change trend, which leads to different life span and predictability of precipitation at different scales [16]. The shorter the life span and the worse the predictability, the smaller the scale of precipitation. If you predict the entire scale of the precipitation field, the prediction error will be too large. The Horn-Schunk method uses global smoothing, while the Lucas-Kanade method uses local matching. In 1995, Laroche et al. proposed a Variational Echo Tracking (VET) algorithm[17]. On this foundation, McGill University in Canada created the MAPLE approach prediction system. Germann et al. used MAPLE to track and predict a 1424h-hour rainfall event in the continental United States, and the results showed that the average forecast time limit based on MAPLE was 5.1h hours, which was clearly better than the forecast effect of the Euler persistence algorithm[18]. Mandapara et al. forecasted 20 precipitation events that occurred in the Swiss alpine region from 2005 to 2010, and the results showed that the time limit of credible forecast based on MAPLE reached 3h in the alpine region[19]. Lee et al. used MAPLE to forecast several summer precipitation events on the Korean Peninsula in 2008, and the results showed that the effective forecasting time was 2.5 hours [20]. In comparison to the life of the precipita-tion model, the Lagrange advection scheme advects the radar precipitation model, which doubles the predictability of a given confidence level.
Given the limitations of traditional forecasting methods in tracking and forecasting, as well as the superiority of the VET algorithm, this study used previous studies, com-bined with the precipitation spectrum decomposition method [21], variational echo tracking technology, and the autoregressive AR2 model [22], to track and forecast three typical rainfalls in Hebei Province during the rainy season, and evaluated the effect of precipitation forecasting.”
Point 5: What is the contribution and novelty of this study? Please clarify them in the introduction and conclusion.
Reply: Thank you for your advice. In this study, we used spectral decomposition algorithms and AR2 on the VET.
To accurately describe the characteristics of precipitation at different scales, spectral decomposition algorithms are used to decompose precipitation or radar echoes. An autoregressive (AR) scheme of forecasting model of order n is used, where n can be 1, 2, 3, and so on . Among them, the second-order AR model (AR2) performs the best and is commonly used.
In this study, these two methods were firstly applied together in the extrapolation method we selected and applied to several rainfall events in Hebei, China.
Point 6: Section 4 Forecasting results is not well written, and it is hard to follow.
Reply: Thank you for your suggestions. The above content has been modified according to the modification suggestions. We further clarified the methods and ways in which we obtained the results. And further clarify the information that Figure 3 wants to present to readers. We have arranged the content expressed in Figures 4 to 6, allowing readers to have an intuitive understanding of our results through the connection between text and images.
“As stated in Principles and Methods, this paper establishes the balance relationship of reflectivity factors of adjacent radar images and obtains the initial echo motion vector field using the variational echo tracing principle and the Lagrange conservation law. Analyze the information characteristics of radar echo images, realize the scale decom-position of the precipitation field, and forecast precipitation independently at each moment. The autoregressive AR2 model was used to forecast extrapolation, and pre-cipitation was superimposed according to scale, resulting in the extrapolation forecast of three selected rainfall events. Figure 3 compares the application of the method to the 0~3h forecast results from 20: 00 on July 21st, 2012.Taking the example of May 22, 2021 at 12:24, as shown in Figure 4 there was no significant change in radar QPE from 12:54 to 13:54, Precipitation is distributed in a band from southwest to northeast. The pre-cipitation began to weaken and move south from 13:54.By 14:24, while the precipitation in the northeastern region region had mostly disappeared, the precipitation in the southern region continued to intensify. Over the 3-h period, the radar QPE showed a pattern of decreasing precipitation in the northeastern region and increasing precipita-tion in the southern region, with a tendency of sorthward movement and scattered precipitation areas. The spatial distribution and intensity of the radar quantitative pre-cipitation forecast (QPF) during 12:54 to 13:24 were similar to the QPE. From 14:24 to15:24, while the precipitation in the sorthward region gradually weakened and almost disappeared by 14:54, the precipitation in the northeastern region gradually intensified, with no significant changes in spatial distribution. The 3-h radar QPF showed a trend of decreasing precipitation in the sorthward region and increasing precipitation in the northeastern region, with precipitation becoming more concentrated in small areas and no scattered precipitation areas. To objectively evaluate the forecasting results, this study conducted evaluations of forecast performance in terms of spatial and temporal scales. The specific evaluation results are as follows:”
Point 7: In lines 459-461 in Conclusion, the authors gave the statement “Overall, this method demonstrates good applicability in the nowcasting of typical precipitation events during the rainy season in Hebei Province. “However, this paper only used three historical heavy rainfall events, and this may be a limitation.
Reply: Thank you for your suggestions. The applicability of this method in Hebei Province can not be explained by the three rainfall events, and the forecast results can only show that the performance is better in the three rainfall events. The content of this paper has been modified. In the later stage, this method will continue to be improved and optimized to achieve more rainfall forecast studies and reach a conclusion with high confidence.
After modification:“Overall, the forecast effect of this method is good in the three rainfall, the highest CSI is up to 0.74, the highest POD is up to 0.87, the forecast accuracy and success rate is high.”
Point 8: English writing must be improved.
Reply: Thank you for your advice. It is true that the English of this paper is not standard. The translation and polishing of this paper were completed by edigate. The content and grammar of this paper have been modified to improve the oral English level. After our modifications, we are still inviting them to further refine.

Reviewer 3 Report
Dear Authors,
In general paper nice to me and after minor corrections I can recommend for publication in the Water Journal. However there are several suggestions.
I suggest a change article title into:
„Application of Variational Optical Flow Forecasting Technique Based on Precipitation Spectral Decomposition to Three Case Studies of Heavy Precipitation Events during Rainy Season in Hebei Province“
At line 40 you mentioned „satellite“ but letter you speak about „Doppler radar echo“. Could you change adopted that.
At line 83 is not clear meaning of „1424h“?
General comment on this type studies that „echo“ image is not precipitation amount, rather „water in the air“ thus please comment this in the paper.
Please make wider figure captions at Fig. 4 and 10.
Author Response
Application of Variational Optical Flow Forecasting Technique Based on Precipitation Spectral Decomposition to Three Case Studies of Heavy Precipitation Events during Rainy Season in Hebei Province
Reply to Reviewer #3
We appreciate the constructive comments from the reviewer. We have modified according to the reviewer's opinion. Thank you for your advice.
Point 1: I suggest a change article title into:
“Application of Variational Optical Flow Forecasting Technique Based on Precipitation Spectral Decomposition to Three Case Studies of Heavy Precipitation Events during Rainy Season in Hebei Province”
Reply: Thank you for your suggestions. The title of the paper before modification is indeed a little broad, but the revised title indicates three typical rainfall events in the study area, which makes the description more straightforward and easy to understand. Thanks again for your comments, the title of the paper has been modified according to the suggestions.
Point 2: At line 40 you mentioned „satellite“ but letter you speak about „Doppler radar echo“. Could you change adopted that.
Reply: Thank you for your advice. Both the satellite mentioned in the paper and the weather radar studied belong to the category of remote sensing. It is my negligence that there is a language error, and the original text has been modified. In addition, the language structure has been adjusted in the introduction part, and the above problems have been solved. Thank you again for your suggestions and guidance.
Point 3: At line 83 is not clear meaning of „1424h“?:
Reply: The paper cited here is “Predictability of Precipitation from Continental Radar Images, Part IV: Limits to Prediction”. And “1424h” refers to 1424h rainfall. As shown in the picture.
Reference:
Germann U, Zawadzki I, Turner B. Predictability of precipitation from continental radar images. Part IV: Limits to prediction[J]. Journal of the Atmospheric Sciences, 2006, 63(8): 2092-2108.
Point 4: General comment on this type studies that „echo“ image is not precipitation amount, rather „water in the air“ thus please comment this in the paper.
Reply: Thank you for your suggestion. Echo image is the result of radar scanning, echo intensity is different from precipitation intensity, but there is a certain conversion relationship. Here, a precipitation inversion method is needed, namely Z-R relation method. Z-R relation is used to convert echo field reflectance into rainfall, Z refers to reflectance factor value, R refers to precipitation intensity, Z is a function of R. The key step of conversion is the determination of parameters. Here, the default values of a and b suggested by the United States WSR-88D radar are adopted. In layered rainfall, a is 200 and b is 1.6. In convective rainfall, a takes 300 and b takes 1.4, which is to realize the conversion between echo and rainfall through this relationship. The description of echo and rainfall in the paper is not accurate enough, so it has been modified. Thanks again for your suggestion.
Point 5: Please make wider figure captions at Fig. 4 and 10.
Reply: Thank you for your suggestion, which has been modified according to your opinions. Figure 4 is a schematic diagram of precipitation spectrum decomposition. Due to the existence of precipitation at different scales in the precipitation field, the life cycle of precipitation at these scales is different. Direct forecast will affect the forecast result. Before the forecast, the precipitation is classified according to the size of the scale and decomposed into different scale layers. The predicted results are accumulated according to the weight. The weight coefficient of long life cycle precipitation is larger, while that of short life cycle precipitation is smaller.
Again, Figure 10 has been supplemented and modified according to the comments. Where, a, b and c are echo motion track of ideal forecast field, and d, e and f are echo motion track of actual forecast field. This figure mainly points out the difference between the ideal forecast field and the actual forecast field, so as to judge the source of error. Thanks again for your advice.
Conclusion words:
We appreciate the constructive comments and valuable suggestions from the reviewer. The manuscript has been carefully revised accordingly. Detailed explanations and discussions are given regarding these issues. In our revision, spelling errors are corrected and efforts are also made to improve the readability of the paper.

Reviewer 4 Report
Major Comments
1) Abstract: The last sentence says, "good applicability", which is quite vague and subjective. Is it possible that the authors provide objective error or reliability confidence percentage?
2) Page 2, lines 44-55: There is no mention about Figure 1 in the text. Besides this is well-known information, which can be mentioned by giving proper citations from the literature,
3) Pages 3 and 4: Figures 2 and 3 are not cited in the text: Furthermore, there is no mention about the past historical or recent flood and flash flood occurrences and occurrence potential in the study area. Unfortunately, these figures are nor described sufficiently, but only exposition of them are given without available depth interpretations related to the purpose of the study. The same is valid for Table 1, which exposes some numbers only without interpretation and relevance to the study purpose,
4) Page 4: All equations are written without proper citations from the literature, as if they are originally proposed by the authors. Inclusion of related citations for each equation is necessary,
5) Page 4, Eq. (3.2): What is the basis of this expression, it seems as continuity (mass balance) equation. The authors should explain its importance and provide relevant citations from the literature. Since the precipitation rate change is equal to zero as in Eq. (3.1), why still in Eqs. (3.2) and (3.3) this term is kept?
6) Page 6: Figure 4 is not interpretatively explained in the text and it is given to the reader as a Picture,
7) Pages 6 and 7: None of the equations is documented by proper references in sub-section "3.3. AR2 Autoregressive Extrapolation Forecasting",
8) Page 9, lines 289-298. There is no need for writing these numbers, because they are already available in Table 3. The authors should provide few important and attractive points from the general interpretation of the numerical results in the table,
9) Page 10, lines 327-336: All these numbers are given in Table 4, therefore their reflections in the text is not necessary. The general outcomes from this table must be interpreted for the purpose of the study,
10) Page 12-14: The figures in "5. Discussion" section, Figure 9 and Figure 10, are not cited in the text; the reader cannot make connection between the figures and the text explained in this section,
11) Page 14: "6. Conclusions" section. The first item (1) is confusing as to the reliability of the proposed methodology,
12) Page 14: The last sentence states again "good applicability" without quantitative comparison with the other existing methodologies.
Author Response
Application of Variational Optical Flow Forecasting Technique Based on Precipitation Spectral Decomposition to Three Case Studies of Heavy Precipitation Events during Rainy Season in Hebei Province
Reply to Reviewer #4
We appreciate the constructive comments from the reviewer. We have modified according to the reviewer's opinion. Thank you for your advice.
Point 1: Abstract: The last sentence says, "good applicability", which is quite vague and subjective. Is it possible that the authors provide objective error or reliability confidence percentage?
Reply: Thank you for your advice. After consideration, it is considered that the forecast results are not enough to prove the applicability of this research method in the whole study area, and can only show that the forecast effect of these three typical rainfall is better. Among the three rainfall forecasts, CSI can reach up to 0.74. The original text has been revised. Thanks again for your suggestion.
After modification::” Overall, the forecast effect of this method is good in the three rainfall, the highest CSI is up to 0.74, the highest POD is up to 0.87, the forecast accuracy and success rate is high.”
Point 2: Page 2, lines 44-55: There is no mention about Figure 1 in the text. Besides this is well-known information, which can be mentioned by giving proper citations from the literature,
Reply: Thank you for your suggestions. According to your suggestions, the figure has been described and explained to some extent in the old paper. After consideration, the introduction part of the article has been reorganized and modified. There may be some missing or added figures, but all of them have been described and explained to some extent. Thank you again for your comments.
After modification: “The occurrence, evolution, and disappearance of precipitation in severe convective weather and precipitation caused by it are very fast, and its prediction and early warning are key and difficult points in the field of meteorology and hydrology. The quality of the observation data, the precipitation prediction method used, and the timeliness of the forecast period all influence the accuracy of its prediction. With the advancement of Doppler weather radar remote sensing technology, high-quality cap-ture of rainfall spatial distribution information and rainfall inversion based on volume scanning model provide basic data support for precipitation forecasting. Based on this, Browning proposed that precipitation approaching forecast is a forecast with high temporal and spatial resolution that the weather will change significantly in a short time (0~3h) by radar echo extrapolation, which has become one of the important researches in the hydrometeorological field [1] [2]. According to Austin and Bellon, the approach prediction algorithm should include two components: echo identification and tracking and extrapolation prediction[3]. The optimal displacement is predicted after the echo is identified and the echo field is established. The approach prediction method based on weather radar echo tracking and extrapolation can show strong, good, and complete convective weather structure and convection movement, especially in areas with con-tinuous radar reflectivity, which can better construct echo field, track, and extrapolate echo displacement changes.”
Point 3: Pages 3 and 4: Figures 2 and 3 are not cited in the text: Furthermore, there is no mention about the past historical or recent flood and flash flood occurrences and occurrence potential in the study area. Unfortunately, these figures are nor described sufficiently, but only exposition of them are given without available depth interpretations related to the purpose of the study. The same is valid for Table 1, which exposes some numbers only without interpretation and relevance to the study purpose.
Reply: Thank you for your suggestion. Figure 2 shows the distribution of radar stations and scanning area in the study area, which has been described in the paper according to modification suggestions. Since the watershed map in Figure 3 has no practical significance in this study, it is deleted. Table 1 describes the time and type of rainfall and other basic information, which has been explained and quoted in the paper according to the modification suggestions. Thank you again for your suggestions.
After modification: “The data used in this study consist of three weather events observed by SA-band Doppler radar within the territory of Hebei Province, China. The radar base data was obtained from the China Meteorological Administration, in **.bin.bz2 format. As shown in Figure 2, the scanning radius of a single S-band radar was 250 km (with an effective scanning radius of 230 km), providing complete coverage of the study area.. The radar operated with a scan completed every 6 min at nine different elevation an-gles, with a spatial resolution of 1 km × 1 km. The three weather events studied were three heavy rainfall events that occurred on July 21, 2012; July 20, 2016; and July 18, 2021 in Hebei Province. The rainfall event on July 21, 2012 was associated with con-vective weather systems, whereas the events on July 20, 2016 and July 18, 2021 were associated with stratiform cloud weather systems,the details are shown in Table 1.”
Point 4: Page 4: All equations are written without proper citations from the literature, as if they are originally proposed by the authors. Inclusion of related citations for each equation is necessary,
Reply: Thank you for your advice. At present, the equation mentioned in the paper has been quoted in relevant literature, which is indeed my negligence. Thank you again for your advice.
Reference:
1)Horn B K P, Schunck B G. Determining optical flow[J]. Artificial intelligence, 1981, 17(1-3): 185-203.
2)Laroche S, Zawadzki I. Retrievals of horizontal winds from single-Doppler clear-air data by methods of cross correlation and variational analysis[J]. Journal of Atmospheric and Oceanic Technology, 1995, 12(4): 721-738.
3)Seed A W. A dynamic and spatial scaling approach to advection forecasting[J]. Journal of Applied Meteorology and Climatology, 2003, 42(3): 381-388.
4)Chang F, Wong A C M. Improved likelihood-based inference for the stationary AR (2) model[J]. Journal of Statistical Planning and Inference, 2010, 140(7): 2099-2110.
Point 5: Page 4, Eq. (3.2): What is the basis of this expression, it seems as continuity (mass balance) equation. The authors should explain its importance and provide relevant citations from the literature. Since the precipitation rate change is equal to zero as in Eq. (3.1), why still in Eqs. (3.2) and (3.3) this term is kept?
Reply: Thank you for your advice. Equation 3.2 is the equilibrium formula of radar reflectance factor at adjacent moments. It is based on Lagrange conservation principle, so the change rate of precipitation rate in unit time is 0. Relevant literature has been modified and quoted. Equations 3.3 and 3.4 are expansions of equations 3.2. Partial derivatives of x, y and t are calculated separately, and their cumulative value is 0. Among them, the partial derivative of x and y will involve the velocity vector, so the formula is retained, and the relevant literature has been modified and quoted.
Reference:
1)Staniforth A, Côté J. Semi-Lagrangian integration schemes for atmospheric models—A review[J]. Monthly weather review, 1991, 119(9): 2206-2223.
2)Horn B K P, Schunck B G. Determining optical flow[J]. Artificial intelligence, 1981, 17(1-3): 185-203.
Point 6: Page 6: Figure 4 is not interpretatively explained in the text and it is given to the reader as a Picture,
Reply: Thanks for your suggestion, the text introduction to Figure 3 has been added according to the modification suggestion. The spectral decomposition of precipitation realizes the division of multi-scale precipitation, and assigns the weight coefficient to the forecast according to the length of the life cycle, which effectively improves the accuracy of the forecast results.
Point 7: Pages 6 and 7: None of the equations is documented by proper references in sub-section "3.3. AR2 Autoregressive Extrapolation Forecasting",
Reply: Thanks for your suggestion, the literature about AR2 model has been cited in the article. AR2 model is a second-order autoregressive model. It uses the relationship between t-1, t-2 and the dependent variable at time t to solve the correlation coefficient, and then establishes the model equation. In precipitation forecast, the dependent variable is the reflectance factor value (unit: DB).
Reference
Chang F, Wong A C M. Improved likelihood-based inference for the stationary AR (2) model[J]. Journal of Statistical Planning and Inference, 2010, 140(7): 2099-2110.
Point 8:Page 9, lines 289-298. There is no need for writing these numbers, because they are already available in Table 3. The authors should provide few important and attractive points from the general interpretation of the numerical results in the table,
Reply: Thank you for your suggestions. The above content has been modified according to the modification suggestions.
Point 9:Page 10, lines 327-336: All these numbers are given in Table 4, therefore their reflections in the text is not necessary. The general outcomes from this table must be interpreted for the purpose of the study,
Reply: Thank you for your suggestions. The above content has been modified according to the modification suggestions. As Point 8 indicated, both two points have been deleted for the repeat of Tables.
Point 10:Page 12-14: The figures in "5. Discussion" section, Figure 9 and Figure 10, are not cited in the text; the reader cannot make connection between the figures and the text explained in this section,
Reply: Thank you for your advice. All the pictures have been mentioned and explained in the article, so readers can connect the pictures with the words. This is my negligence, thanks again for your advice.
Point 11:Page 14: "6. Conclusions" section. The first item (1) is confusing as to the reliability of the proposed methodology,
Reply: Thank you for your suggestion. The variable spectral flow extrapolation technology based on precipitation spectral decomposition is a precipitation forecast research carried out on the basis of precipitation spectral decomposition, combined with variable spectral flow technology and AR2 model extrapolation technology. These three technologies have certain research basis. At the same time, the prediction results of radar QPF and radar QPE have been completed accuracy evaluation, the results show that the forecast effect of this method is good in the three rainfall, the highest CSI is up to 0.74, the highest POD is up to 0.87, the forecast accuracy and success rate is high. Thanks again for your advice.
Reference:
1)Laroche S, Zawadzki I. Retrievals of horizontal winds from single-Doppler clear-air data by methods of cross correlation and variational analysis[J]. Journal of Atmospheric and Oceanic Technology, 1995, 12(4): 721-738.
2)Seed A W. A dynamic and spatial scaling approach to advection forecasting[J]. Journal of Applied Meteorology and Climatology, 2003, 42(3): 381-388.
3)Chang F, Wong A C M. Improved likelihood-based inference for the stationary AR (2) model[J]. Journal of Statistical Planning and Inference, 2010, 140(7): 2099-2110.
Point 12:Page 14: The last sentence states again "good applicability" without quantitative comparison with the other existing methodologies.
Reply: Thanks for your suggestion. According to the forecast results, the variable spectral flow method based on precipitation spectral decomposition has a good forecast effect on the three rainfall, in which the highest CSI is up to 0.74 and the highest POD is up to 0.81, but it is not enough to show that it has good applicability in the whole study area. The content of the paper has been revised. At the same time, the comparison with other methods is added in the discussion chapter and the conclusion chapter, and the effect evaluation of the method is completed.

Round 2
Reviewer 2 Report
Point 3: Generally, the reference number should be at the end of the sentence.
Reply: Thank you for your guidance. According to the revision opinions, the current reference numbers are all put at the end of the sentence, which is our negligence. Thank you again for your suggestion.
Response: I do not think the authors figured out this issue in the revised manuscript. Please check all the references.
Also, the revised manuscript in the downloaded PDF version has a reference format issue. In the revised text, I only see references 4-9 are given the brackets, all the other references are given the number directly. For example, line 49, “echo identification and tracking and extrapolation prediction3.” For another example, in line 88, Laroche et al. proposed a Variational Echo Tracking (VET) algorithm17. ” Numbers should be placed in square brackets [ ] and uniformed.
Many sentences are very confusing. For example, in lines 128-130, “The Lagrangian conservation algorithm is currently the most commonly used method for advection estimation24, Optical flow method is based on this principle for echo tracking25”, and in line 155, ”the calculation formula is as follows26:” do 24, 25, and 26 mean reference numbers?
Please check all the references.
Minor editing of the English language required
Author Response
Point 3: Generally, the reference number should be at the end of the sentence.
Reply: Thank you for your guidance. According to the revision opinions, the current reference numbers are all put at the end of the sentence, which is our negligence. Thank you again for your suggestion.
Response: I do not think the authors figured out this issue in the revised manuscript. Please check all the references.
Also, the revised manuscript in the downloaded PDF version has a reference format issue. In the revised text, I only see references 4-9 are given the brackets, all the other references are given the number directly. For example, line 49, “echo identification and tracking and extrapolation prediction3.” For another example, in line 88, Laroche et al. proposed a Variational Echo Tracking (VET) algorithm17. ” Numbers should be placed in square brackets [ ] and uniformed.
Reply:
Thank you for your suggestion. It was an oversight on our part that the [] of the reference number disappeared during the layout process. We have now put the reference number at the end of the sentence and put the number in the [] as you suggested, and thank you again for your suggestion.
Many sentences are very confusing. For example, in lines 128-130, “The Lagrangian conservation algorithm is currently the most commonly used method for advection estimation24, Optical flow method is based on this principle for echo tracking25”, and in line 155, ”the calculation formula is as follows26:” do 24, 25, and 26 mean reference numbers?
Reply:
Thank you for your suggestions. The [] of the reference number disappeared during the layout process, which is an oversight on our part. We have now put the reference number at the end of the sentence and put the number in the []. 24, 25, 26 refer to the reference number. We are sorry for the confusion caused to you, and we will revise the reference number in the whole text according to the revision.
Please check all the references.
Reply:
Thank you for your suggestions. The failure to place the literature numbers in [] was a major oversight on our part, and the literature numbers in the full text have now been re-corrected as requested. We apologize for the inconvenience caused and thank you for your advice and guidance.
In addition, in order to express the meaning of the paper more clearly, we have improved our English writing and made some minor adjustments to the sentences in the paper, as follows:
- Line 19,”triggering” to “trigger”
- Line 22,”apply” to “applied”
- Line 27,”weakens” to “weakened”
- Line 27,”occurs” to “occured”
- Line 30,”is up to” to “reached”
- Lines 31,32,33,” Overall, the forecast effect of this method is good in the three 31 rainfall, the highest CSI is up to 0.74, the highest POD is up to 0.87, the forecast accuracy and success 32 rate is high.” to “Overall, this method provided an effective forecast for the three rainfall events, with the highest CSI of 0.74 and highest POD of 0.87, and a high forecast accuracy and success rate.”
- Lines 24,109, 115, 119,437, Hebei Province to The Hebei Province.
- Lines 112,117,173,244,280,339,342,345,351,354,357,381,398, at the end of a sentence involving a reference picture or reference table, the form (Figure ...) is used instead of as shown in figure ... or (Table ...) instead of as shown in figure ... and as shown in table ...
- Lines 112,113,“The radar operated with a scan completed every 6 min at nine different elevation angles, with a spatial resolution of 1 km × 1 km” to “The radar completed a 112 scan every 6 min at nine different elevation angles, with a spatial resolution of 1 × 1 km.”
- Line 123,”or”to “and”
- Line 124,”determines”to”determine”
- Line 130,” Optical flow”to” The Optical flow”
- Line 158,” They are defined as follows:”to” The components are de- 158 fined as follows:”
- Line 189,190, the images abc are explained separately
- Line 197, Deleted “Among them”
- Line 198,A most was added before the commonly
- Line 239,An and was added before “critical success index”
- Line 240,”spatial and temporal scales”. to “spatio temporal scales”
- Lines 242,243, “needs to set a threshold TR, judge the relationship between reflectance factor and TR” to “requires a threshold TR, to then judge the relationship between reflectance factor and TR”
- Line 244,”get” to “obtain”
- Lines 249,251,” NC represents cases”to “NC is”
- Line 280, “Taking the example of”to” Considering … as an example
- Line 281,282,a “with” was added before “Precipitation is distributed”
- Lines 287,289,292, “sorthward”to”southward”
- Line 301,”indicates”to “indicate”
- Lines 303-315, Verbs into the past tense
- Lines 320,321,322,323,324,326,327,”is”to”was”, Verbs into the past tense
- Line 334,” clearly superior to rainfall events I and III.”to “clearly superior to the forecast of rainfall events I and III.”
- Line 339,344,”starts to below”to”dropped”
- Line 343,”2 to 3 h compared to 1 - 2 h”to “2 - 3 h compared to that between 1 - 2 h”
- Lines 337-360, Verbs into the past tense
- Lines 386,387,exhibits to exhibited;”decreases” to ”decreased”
- Line 389,” indicating that certain limitations still occur in this method.” to ” indicating that this method still presents certain limitations”
- Lines 404,405,” compared with Kanade-Lucas-Tomasi optical flow method” to “Compared with that of the cross-correlation method”
- Line 407, “echo” to “echoes”
- Line 411,”under“ to ”require”
- Line 414,” causation”to “cause”
- Line 426,”in” to “for”
- Line 439,” AR2” to “ the AR2”
- Line 440,” The forecasting” to “The precipitation forecasting”
- Line 440,441,” was evaluated in terms of spatial and temporal scales.” to “evaluated according to spatio temporal scales”
- Line 446,”and” to “for which”
- Line 453,”Ⅱ” to “Ⅲ”
- Lines 457-461,” Overall, the forecast effect of this method is good in the three rainfall, the highest CSI is up to 0.74, the highest POD is up to 0.87, the forecast accuracy and success rate is high., but there are still some deviations. The research will continue in the later period, and strive to find a better forecasting method.” to “Overall, this method has an effective forecast for the three rainfall events,with the highest CSI of up to 0.74 and ,highest POD of 0.87, and a high forecast accuracy and success rate, there are still some deviations. This research will be continue in an effort to further improve the better forecasting method.”

Reviewer 4 Report
In the text there are numbers as references. THsy should be shown inside a big bracket like [1] and so on
Author Response
Point1:In the text there are numbers as references. THsy should be shown inside a big bracket like [1] and so on
Reply:Thank you for your suggestions. The [] of the reference number disappeared during the layout process, which was an oversight on our part. We have now corrected the literature numbering in the full text according to the revision, and we apologize for the inconvenience caused to you. Thank you again for your suggestions and guidance.
In addition, in order to express the meaning of the paper more clearly, we have improved our English writing and made some minor adjustments to the sentences in the paper, as follows:
- Line 19,”triggering” to “trigger”
- Line 22,”apply” to “applied”
- Line 27,”weakens” to “weakened”
- Line 27,”occurs” to “occured”
- Line 30,”is up to” to “reached”
- Lines 31,32,33,” Overall, the forecast effect of this method is good in the three 31 rainfall, the highest CSI is up to 0.74, the highest POD is up to 0.87, the forecast accuracy and success 32 rate is high.” to “Overall, this method provided an effective forecast for the three rainfall events, with the highest CSI of 0.74 and highest POD of 0.87, and a high forecast accuracy and success rate.”
- Lines 24,109, 115, 119,437, Hebei Province to The Hebei Province.
- Lines 112,117,173,244,280,339,342,345,351,354,357,381,398, at the end of a sentence involving a reference picture or reference table, the form (Figure ...) is used instead of as shown in figure ... or (Table ...) instead of as shown in figure ... and as shown in table ...
- Lines 112,113,“The radar operated with a scan completed every 6 min at nine different elevation angles, with a spatial resolution of 1 km × 1 km” to “The radar completed a 112 scan every 6 min at nine different elevation angles, with a spatial resolution of 1 × 1 km.”
- Line 123,”or”to “and”
- Line 124,”determines”to”determine”
- Line 130,” Optical flow”to” The Optical flow”
- Line 158,” They are defined as follows:”to” The components are de- 158 fined as follows:”
- Line 189,190, the images abc are explained separately
- Line 197, Deleted “Among them”
- Line 198,A most was added before the commonly
- Line 239,An and was added before “critical success index”
- Line 240,”spatial and temporal scales”. to “spatio temporal scales”
- Lines 242,243, “needs to set a threshold TR, judge the relationship between reflectance factor and TR” to “requires a threshold TR, to then judge the relationship between reflectance factor and TR”
- Line 244,”get” to “obtain”
- Lines 249,251,” NC represents cases”to “NC is”
- Line 280, “Taking the example of”to” Considering … as an example
- Line 281,282,a “with” was added before “Precipitation is distributed”
- Lines 287,289,292, “sorthward”to”southward”
- Line 301,”indicates”to “indicate”
- Lines 303-315, Verbs into the past tense
- Lines 320,321,322,323,324,326,327,”is”to”was”, Verbs into the past tense
- Line 334,” clearly superior to rainfall events I and III.”to “clearly superior to the forecast of rainfall events I and III.”
- Line 339,344,”starts to below”to”dropped”
- Line 343,”2 to 3 h compared to 1 - 2 h”to “2 - 3 h compared to that between 1 - 2 h”
- Lines 337-360, Verbs into the past tense
- Lines 386,387,exhibits to exhibited;”decreases” to ”decreased”
- Line 389,” indicating that certain limitations still occur in this method.” to ” indicating that this method still presents certain limitations”
- Lines 404,405,” compared with Kanade-Lucas-Tomasi optical flow method” to “Compared with that of the cross-correlation method”
- Line 407, “echo” to “echoes”
- Line 411,”under“ to ”require”
- Line 414,” causation”to “cause”
- Line 426,”in” to “for”
- Line 439,” AR2” to “ the AR2”
- Line 440,” The forecasting” to “The precipitation forecasting”
- Line 440,441,” was evaluated in terms of spatial and temporal scales.” to “evaluated according to spatio temporal scales”
- Line 446,”and” to “for which”
- Line 453,”Ⅱ” to “Ⅲ”
- Lines 457-461,” Overall, the forecast effect of this method is good in the three rainfall, the highest CSI is up to 0.74, the highest POD is up to 0.87, the forecast accuracy and success rate is high., but there are still some deviations. The research will continue in the later period, and strive to find a better forecasting method.” to “Overall, this method has an effective forecast for the three rainfall events,with the highest CSI of up to 0.74 and ,highest POD of 0.87, and a high forecast accuracy and success rate, there are still some deviations. This research will be continue in an effort to further improve the better forecasting method.”
